# Child Welfare Investigations of Exposure to Intimate Partner Violence Referred by Medical Professionals in Ontario: A Uniquely Vulnerable Population?

**DOI:** 10.3390/healthcare11182599

**Published:** 2023-09-21

**Authors:** Nicolette Joh-Carnella, Eliza Livingston, Jill Stoddart, Barbara Fallon

**Affiliations:** Factor-Inwentash Faculty of Social Work, University of Toronto, Toronto, ON M5S 1V4, Canada; eliza.livingston@utoronto.ca (E.L.); jill.stoddart@utoronto.ca (J.S.); barbara.fallon@utoronto.ca (B.F.)

**Keywords:** child welfare, intimate partner violence, healthcare, policy

## Abstract

Victims of intimate partner violence (IPV) and their children may be at an increased risk for negative health outcomes and may present to healthcare settings. The objective of the current study is to examine the profile of medical-referred child welfare investigations of exposure to IPV in Ontario, Canada. Data from the Ontario Incidence Study of Reported Child Abuse and Neglect 2018 were used. We compared medical-referred investigations with all other investigations of exposure to IPV. Descriptive and bivariate analyses as well as a logistic regression predicting transfers to ongoing services were conducted. Six percent of investigations of exposure to IPV conducted in Ontario in 2018 were referred by a medical source. Compared to other investigations of exposure to IPV, these investigations were more likely to involve younger children (*p* = 0.005), caregivers with mental health issues (*p* < 0.001) and few social supports (*p* = 0.004), and households noted to be overcrowded (*p* = 0.001). After controlling for clinical case characteristics, investigations of exposure to IPV referred by healthcare sources were 3.452 times as likely to be kept open for ongoing child welfare services compared to those referred by other sources (95% CI [2.024, 5.886]; *p* < 0.001). Children and their families who are identified in healthcare settings for concerns of exposure to IPV tend to receive extended child welfare intervention compared to those identified elsewhere. There is a clear difference in service provision in healthcare-originating investigations of exposure to IPV versus investigations originating from other sources. Further research into the services provided to victims of IPV and their children is needed.

## 1. Introduction

In Canada, concerns regarding child well-being and safety can be reported to local child welfare agencies who determine the need for intervention. Studies have investigated the rate and characteristics of investigated child maltreatment in Canada at the national and provincial levels, allowing for the development of evidence-informed policies and practices [1].

Canadian child welfare systems frequently investigate concerns related to children’s exposure to intimate partner violence (IPV) [1]. IPV, as defined by the World Health Organization, includes “any behaviour within an intimate relationship that causes physical, psychological or sexual harm to those in the relationship” [2]. Compared to other forms of investigated maltreatment (i.e., physical abuse, sexual abuse, neglect, and emotional maltreatment), exposure to IPV investigations comprise the largest proportion of substantiated child maltreatment investigations in Canada and Ontario [1,3]. The high rate of investigations of substantiated exposure to IPV is unique within the Canadian child welfare system; exposure to IPV is not a type of investigation that is routinely reported in Australian or American child welfare data [1]. Canadian studies have found that investigations of exposure to IPV tend to involve younger children, connection to support services that are external to child welfare, and substantiation of maltreatment without the provision of ongoing child welfare services or placement in out-of-home care [4,5,6]. It could be that child welfare represents an essential source of support and connection to necessary services for this vulnerable population. On the other hand, the tendency to substantiate exposure to IPV without the provision of ongoing child welfare services might suggest a limitation to the extent of support that child welfare can offer in these investigations.

Negative effects on children’s mental and physical health associated with exposure to IPV have been reported [7,8,9]. As caregivers who experience IPV may also have negative health outcomes, including acute injuries, healthcare professionals represent important points of contact for children and families and need to be aware of the signs and how to screen for potential exposure to IPV [10,11]. Data from Ontario, Canada indicate that most child welfare investigations of exposure to IPV are the result of police reports, and therefore, reports by healthcare professionals represent a relatively understudied area [12]. 

Child welfare in Canada is legislated at the provincial/territorial levels [1]. In Ontario, child welfare services are mandated by the Child, Youth and Family Services Act [13]. Concerns of potential child maltreatment are directed to local Children’s Aid Societies or Child and Family Service Agencies, which are funded by Ontario’s Ministry of Children, Community and Social Services but operate as private non-profit organizations [3]. As defined by the Child, Youth and Family Services Act, there is a mandate in Ontario to report situations in which a child is at risk of or has suffered harm due to the actions or inactions of a caregiver [13]. While this legislation does not directly mandate the report of instances of an exposure to IPV, the assessment tool used in screening for child welfare investigations in the province, known as the Eligibility Spectrum, interprets this legislation to include violence between caregiver(s) or the child’s caregiver and their partner [14]. Following a report to an Ontario child welfare organization, screening workers at the agency use the Eligibility Spectrum to determine whether the concerns meet the threshold to be opened for investigation.

In this study, we examine the profile of investigations of exposure to IPV referred to child welfare from healthcare settings. Our previous study examining the characteristics of child maltreatment-related investigations reported by healthcare professionals in Ontario found that 29% and 22% of investigations reported by hospital-based and community-based healthcare providers, respectively, involved a primary caregiver who was a victim of IPV [15]. Due to the physical and mental health impacts of IPV, caregivers and their children may require medical care, either in the emergency department or through primary care, depending on the urgency of their needs [11,16]. Previous studies demonstrate that victims of IPV utilize healthcare services more than those who have not experienced IPV [11,17,18]. It is estimated that there were over 10,000 emergency department visits related to domestic violence between 2012 and 2016 in Ontario, Canada [19]. Not only are victims of IPV more likely to access health services, but so are their children who may have been exposed to the violence [8,11,20,21,22]. 

Overall, it is well documented that there may be negative health consequences associated with being a victim of or being exposed to IPV. We are not aware of any studies to date that have looked at the profile of child welfare-involved families identified in healthcare settings for concerns of exposure to IPV or the child welfare response in these investigations. Our hypothesis is that medical-referred investigations of exposure to IPV will involve younger children and increased caregiver/household risk factors compared to investigations of exposure to IPV referred by other sources. We further expect medical-referred investigations of exposure to IPV to be more likely to be kept open for ongoing services given that the majority of investigations of exposure to IPV in Ontario are referred by police and tend to be substantiated without being transferred to ongoing services [6,12].

In order to test these hypotheses, our objective is to examine child maltreatment investigations of exposure to IPV in which the report of alleged child maltreatment came from a medical source. Data from the Ontario Incidence Study of Reported Child Abuse and Neglect 2018 (OIS-2018) are used. The medical referral sources captured in the OIS-2018 include hospital personnel, community physicians, and community health nurses. The specific research questions investigated in this study are the following: 

What percentage of investigations of exposure to IPV were referred by medical personnel?Compared to investigations of exposure to IPV referred by other sources, what is the profile of these investigations referred by medical sources? Controlling for clinical and case characteristics, is a medical referral source associated with the provision of ongoing child welfare services in investigations of exposure to IPV? 

## 2. Materials and Methods

### 2.1. Sampling and Weighting

Secondary analysis of data from the OIS-2018 was conducted to answer the research questions. The OIS-2018 is the sixth provincial-level study investigating the incidence of child maltreatment-related investigations carried out in Ontario, which is Canada’s most populous province. Ethics approval for this study was obtained from the Health Sciences Research Ethics Board of the University of Toronto.

The OIS-2018 used a case review methodology in which child welfare workers provided information on investigations they conducted by completing online instruments. Investigations are the result of reports to child welfare organizations that meet the threshold for investigation based on the screening tool used in the province, the Eligibility Spectrum [10]. The OIS-2018 employed a multi-stage sampling design to obtain a representative sample of child welfare investigations conducted in Ontario in 2018. First, a sample of 18 (from a total 48) child welfare organizations in Ontario was selected for participation; consent for study participation was obtained at the organization level. The sampling period included investigations opened between 1 October and 31 December 2018. At larger agencies, the number of investigations included in the study was capped at 250. Lastly, participating workers identified children who were investigated for maltreatment-related concerns within selected cases. The final sample included 7590 child-maltreatment-related investigations involving children 0–17 years old in Ontario. These data were then weighted to provide an annualized provincial estimate. Please see [3] for a description of the study’s weighting procedures. The final weighted estimate for the OIS-2018 was 158,476 investigations involving children 0–17 years old. 

### 2.2. Data Collection Instrument

The online instrument completed by participating child welfare workers included three sections: (1) Intake Information, (2) Household Information, and (3) Child Information. The Intake Information section asked workers to include information on the referral, type of investigation conducted, and the household composition. The Household Information section included questions regarding caregivers living in the home, any potential household risk factors, previous child welfare investigations, transfers to ongoing services, and referrals to non-child-welfare-related services. The Child Information section collected information on child characteristics, forms and severity of maltreatment, and outcomes of investigations.

The OIS-2018 definition of child maltreatment-related investigations included investigations assessing allegations of maltreatment as well as those in which there were no specific allegations of maltreatment, but rather, the risk of future maltreatment for the child was being investigated. Where workers identified their investigation to be focused on allegations of potential maltreatment, they could indicate one of five primary forms of maltreatment: physical abuse, sexual abuse, neglect, emotional maltreatment, and exposure to IPV. Secondary and tertiary forms of maltreatment could also be noted.

The OIS-2018 data collection instrument asked about the individual(s) who made the report to the child welfare agency that resulted in the sampled investigation. The following referral sources were included in the OIS-2018: custodial parent, non-custodial parent, child, relative, neighbour/friend, social assistance worker, crisis service/shelter, community/recreation centre, hospital (any personnel), community health nurse, community physician, community mental health professional, school, other child welfare service, daycare centre, police, community agency, anonymous, and other. Multiple referral sources could be noted. In the present paper, we define medical-referred investigations as those with at least one referral from hospital personnel, a community health nurse, or a community physician. 

### 2.3. Data Analysis

SPSS Statistics version 28 was used to conduct the present analysis. Using the weighted estimate of investigations of exposure to IPV, descriptive and bivariate statistics examining investigations referred by medical sources vs. other referral sources were conducted. Please see Appendix A for a summary of the variables used in the bivariate analyses. Chi-squared tests of significance were conducted using the sample weight for the OIS-2018. The sample weight weighs the estimate back down to the sample size in order to adjust for inflation of the chi-squared statistic due to the size of the estimate. 

A logistic regression predicting transfers to ongoing child welfare services in investigations of exposure to IPV was conducted using the sample weight. Predictors were entered into the model in five blocks and were determined using chi-squared tests of significance, comparing various predictors with respect to transfers to ongoing services. The first block of the model included child ethnicity/race/Indigeneity and at least one functioning concern in the child noted by the investigating worker. The second block included the following caregiver risk factors: alcohol abuse, mental health issues, and few social supports. Household risk factors including overcrowding, two or more moves in the past year, and running out of money for basic necessities in the past six months were included in the third block. The fourth block included previous child welfare investigations and emotional harm noted to the child. The final block included our variable of interest, medical referral sources. 

## 3. Results

Of the estimated 29,028 investigations of exposure to IPV captured in the OIS-2018 (10.82 investigations per 1000 children in Ontario), six percent (an estimated 1699 investigations; 0.63 investigations per 1000 children in Ontario) were referred by medical personnel (see Table 1). Table 2 compares investigations of exposure to IPV referred by medical professionals and all other referral sources according to the child, caregiver, household, and case characteristics. The medical-referred investigations of exposure to IPV involved younger children, with 43% of these investigations involving children 0–3 years old (compared to 25% of investigations of exposure to IPV referred by other sources; *p* = 0.005; see Table 2). 

Primary caregivers in investigations of exposure to IPV referred by medical professionals were more likely to have mental health issues identified by investigating workers (38% of these investigations) compared to investigations referred by other sources (22% of these investigations; *p* < 0.001; see Table 2). Primary caregivers in medical-referred investigations of exposure to IPV were also more likely to have few social supports noted by the investigating child welfare workers (34% of investigations referred by medical professionals vs. 21% of investigations referred by all other referral sources; *p* = 0.004; see Table 2). Ten percent of the investigations of exposure to IPV referred by medical professionals were noted to involve households that were assessed by the investigating child welfare workers to be overcrowded compared to three percent of all other investigations of exposure to IPV (*p* = 0.001; see Table 2). 

Emotional harm to the child as a result of substantiated maltreatment was significantly less likely to be noted in investigations of exposure to IPV referred by medical personnel compared to all other investigations of exposure to IPV (noted in 8% of investigations of exposure to IPV referred by medical professionals and 24% of investigations of exposure to IPV referred by other sources; *p* = 0.001; see Table 2). Lastly, 40% of the investigations of exposure to IPV referred by medical professionals were kept open for ongoing child welfare services compared to only 22% of all other investigations of exposure to IPV (*p* < 0.001; see Table 2). 

A logistic regression predicting transfers to ongoing child welfare services is presented in Table 3. As demonstrated in this table, various child (ethnicity/race/Indigeneity and at least one functioning concern), primary caregiver (noted alcohol abuse, mental health issues, and few social supports), household (overcrowding and running out of money for basic necessities), and case (emotional harm noted to the child) characteristics were significant predictors of the decision to transfer a case to ongoing services. After controlling for other variables in the model, investigations of exposure to IPV referred by medical professionals were significantly more likely to be transferred to ongoing services compared to all other investigations of exposure to IPV (odds ratio = 3.452; 95% CI [2.024, 5.886]; *p* < 0.001). 

## 4. Discussion

As a result of increased vulnerability to mental and physical health concerns for both children, who are exposed to violence, and their caregivers, who are victims of violence, children who are exposed to IPV may be more likely to come to the attention of medical professionals compared to their peers [8,11,20,21,22]. As such, medical professionals serve important roles in the identification of suspected exposure to IPV that may require child welfare intervention. The purpose of our study was to establish the proportion of investigations of exposure to IPV referred by medical sources, describe the profile of these investigations, and determine if medical referrals for investigations of exposure to IPV were associated with ongoing child welfare service provision when controlling for other factors.

### 4.1. Profile of Medical-Referred Investigations of Exposure to IPV 

The results of our bivariate analyses indicate an increased proportion of certain risk factors in medical-referred investigations of exposure to IPV compared to those referred by other sources. Consistent with previous studies examining trends in hospital-based and medical referrals to child welfare agencies in Ontario, investigations of exposure to IPV referred by medical personnel involved younger children [23] and were more likely to involve primary caregivers who were noted to struggle with mental health issues and few social supports [15]. These investigations were also more likely to involve households that were noted to be overcrowded.

### 4.2. Ongoing Child Welfare Services in Medical-Referred Investigations of Exposure to IPV 

The results of our multivariate analysis reveal that, when controlling for clinical case characteristics, the investigations of exposure to IPV referred by healthcare professionals were nearly three and a half times as likely to be transferred to ongoing child welfare services compared to those referred by other sources (see Table 3). It could be that the increased vulnerability of this population (as evidenced by the young ages of the investigated children and the presence of certain caregiver and household risk factors) heightens child welfare workers’ concerns for the overall well-being of these children. Therefore, these investigations are kept open to help establish more support for these families. Interestingly, these investigations were less likely to involve emotional harm to the child. This is, again, consistent with the workers keeping these investigations open to support the families and help to mitigate risk factors rather than to protect children from emotional harm as a result of exposure to IPV. 

Previous work examining medical referrals for child-maltreatment-related concerns demonstrated that these investigations were more likely to be substantiated and involve more intrusive forms of child welfare involvement compared to investigations referred by other sources [15,23]. In a recent qualitative study examining the intersection of the child welfare and healthcare systems, child welfare workers reflected that this could be due to the perceived expertise and credibility of healthcare providers as well as the increased severity of cases referred by these sources [24]. Although there was no significant difference in the substantiation between medical-referred investigations of exposure to IPV and those referred by other sources, these reasons could contribute to the increased likelihood of investigations referred by medical personnel being transferred to ongoing services. 

A study using data from the Canadian Incidence Study of Reported Child Abuse and Neglect 2003 documented a propensity for investigations of exposure to IPV to involve substantiated maltreatment but not be transferred for ongoing child welfare services or to be placed in out-of-home care [6]. More than half of the investigations of exposure to IPV in Ontario are referred by police [12]. Nikolova et al. [5] conducted interviews with representatives from police departments in Ontario to investigate the recent increase in investigations of exposure to IPV in the province. These representatives described a mandatory reporting policy requiring police to report all potential instances of exposure to verbal, emotional, or physical violence to child welfare agencies, even if the child is not physically present [5].

As previously mentioned, the screening tool used by child welfare agencies in the province, the Eligibility Spectrum, defines exposure to IPV as a reason for child welfare investigation. Therefore, police calls for exposure to IPV essentially automatically result in child welfare investigations. However, the police do not necessarily make these referrals based on clinical concern for the child, but rather due to the presence of a child. This is different from referrals by medical personnel where clinicians are likely making the decision to refer to child welfare based on the suspicion of risk or harm to the child. This could help to explain why investigations referred by medical personnel are more likely to be transferred to ongoing child welfare services. 

### 4.3. Limitations

Several limitations should be considered when interpreting the findings of the current study. First, the data collected in the OIS-2018 are cross-sectional and represent child welfare workers’ knowledge at the conclusion of their initial investigations. These data represent the clinical judgements of the workers and are not independently verified. As the OIS-2018 only captures investigations, child maltreatment cases that are not reported to child welfare, investigated only by police, or screened out prior to investigation are not included. Furthermore, no information regarding dispositions after the investigation stage of child welfare involvement is included. Three limitations to the weighting procedures should be noted. The correction applied to account for the agency size uses the overall service volume but does not consider the variation in investigation types across agencies. The annualization correction only accounts for seasonal fluctuations in the number of investigations but not the types of investigations conducted. Finally, cases that were re-opened within the same year are included in the annualization calculation, meaning multiple investigations of the same child can be counted. For this reason, child-level investigations is the unit of analysis of the OIS-2018 rather than investigated children. 

## 5. Conclusions

Healthcare professionals are important points of contact for potential victims of IPV and their children who may be exposed to violence. Six percent of all investigation referrals to Ontario child welfare agencies for exposure to IPV originate from a healthcare source. Supporting our initial hypotheses, compared to investigations of exposure to IPV referred by other sources, medical-referred investigations were more likely to involve younger children and several caregiver and household risk factors, indicating a uniquely vulnerable population. Medical-referred child welfare investigations were also more likely to be kept open for ongoing child welfare services following the initial investigation. Postulated reasons for this include the aforementioned increased vulnerability of the children/families, perceived expertise of the healthcare professionals, or the nature of police-referred investigations, which represent the majority of investigations of exposure to IPV. Further research into the services provided to victims of IPV and their children, identified in healthcare settings, would help to elucidate how resources can be directed to identify and serve these families’ needs. 

## Figures and Tables

**Table 1 healthcare-11-02599-t001:** Medical referral sources vs. all other referral sources in investigations of exposure to IPV in Ontario in 2018.

Referral Source	#	Rate per 1000	%
Medical referral source	1699	0.63	6%
All other referral sources	27,329	10.19	94%
Total investigations of exposure to IPV	29,028	10.82	100%

Based on unweighted sample of 1392 investigations of exposure to IPV.

**Table 2 healthcare-11-02599-t002:** Characteristics of medical referral sources vs. all other referral sources in investigations of exposure to IPV in Ontario in 2018.

	Medical Referral Sources			Other Referral Sources			Total IPV Investigations			
	#	Rate per 1000	%	#	Rate per 1000	%	#	Rate per 1000	%	*p*-Value *
Child factors										
Age										0.005
0–3 years	724	0.27	43%	6781	2.53	25%	7506	2.80	26%	
4–7 years	372	0.14	22%	8025	2.99	29%	8397	3.13	29%	
8–11 years	340	0.13	20%	7321	2.73	27%	7661	2.86	26%	
12–17 years	262	0.10	15%	5201	1.94	19%	5463	2.04	19%	
										0.002
White	586	0.22	34%	13,623	5.08	50%	14,209	5.30	49%	
Black	306	0.11	18%	3631	1.35	13%	3937	1.47	14%	
Indigenous	133	0.05	8%	2566	0.96	9%	2699	1.01	9%	
Latin American	0	0.00	0%	1062	0.40	4%	1062	0.40	4%	
Other	673	0.25	40%	6447	2.40	24%	7120	2.65	25%	
Child functioning concerns								0.00	0%	
Developmental/physical	228	0.09	13%	2782	1.04	10%	3010	1.12	10%	0.327
Emotional	107	0.04	6%	2797	1.04	10%	2904	1.08	10%	0.237
Behavioural	121	0.05	7%	2093	0.78	8%	2215	0.83	8%	0.915
Primary caregiver factors										
Age										0.843
<22	-	-	3%	590	0.22	2%	639	0.24	2%	
>21	1650	0.62	97%	26,739	9.97	98%	28,389	10.59	98%	
Not cooperative/not contacted	-	-	3%	1721	0.64	6%	1777	0.66	6%	-
Alcohol abuse	-	-	5%	2935	1.09	11%	3025	1.13	10%	-
Drug abuse	178	0.07	10%	1784	0.67	7%	1961	0.73	7%	0.117
Mental health issues	640	0.24	38%	5895	2.20	22%	6536	2.44	23%	<0.001
Few social supports	576	0.21	34%	5636	2.10	21%	6211	2.32	21%	0.004
Household factors										
Income source										0.69
Full-time	885	0.33	52%	15,226	5.68	56%	16,112	6.01	56%	
Part-time	150	0.06	9%	3268	1.22	12%	3417	1.27	12%	
Other benefits	438	0.16	26%	5729	2.14	21%	6167	2.30	21%	
Unknown	-	-	5%	1273	0.47	5%	1363	0.51	5%	
None	136	0.05	8%	1832	0.68	7%	1968	0.73	7%	
2 or more moves in the past year	-	-	6%	1356	0.51	5%	1453	0.54	5%	-
Overcrowded home	170	0.06	10%	858	0.32	3%	1028	0.38	4%	0.001
Ran out of money for basic necessities in the past 6 months	135	0.05	8%	2893	1.08	11%	3027	1.13	10%	0.359
Child harm										
Emotional harm	136	0.05	8%	6581	2.45	24%	6718	2.51	23%	0.001
Physical harm	-	-	4%	221	0.08	1%	281	0.10	1%	-
Child welfare involvement										
Previous investigation	1045	0.39	62%	18,630	6.95	68%	19,675	7.34	68%	0.231
Substantiation	998	0.37	59%	16,551	6.17	61%	17,550	6.54	60%	0.713
Transfers to ongoing services	684	0.26	40%	6141	2.29	22%	6826	2.55	24%	<0.001
Referrals to services	874	0.33	51%	12,214	4.55	45%	13,088	4.88	45%	0.249
Out-of-home placement	-	-	4%	235	0.09	1%	302	0.11	1%	-

* *p*-value compares medical referrals and all other referral sources for investigations of exposure to IPV. —Estimate was <100 investigations. Based on unweighted sample of 1392 investigations of exposure to IPV.

**Table 3 healthcare-11-02599-t003:** Logistic regression predicting transfers to ongoing child welfare services in Ontario in 2018.

Variables	B	SE	*p*-Value	Odds Ratio	95% Confidence Interval	
Child Factors						
(White as reference)						
Black	−0.330	0.239	0.166	0.719	0.450	1.147
Indigenous	1.061	0.225	<0.001	2.890	1.858	4.495
Latin American	−0.751	0.482	0.119	0.472	0.183	1.213
Other	−0.606	0.198	0.002	0.546	0.370	0.804
At least one functioning concern	0.443	0.167	0.008	1.557	1.123	2.158
Primary Caregiver Factors						
Alcohol abuse	0.777	0.217	<0.001	2.175	1.422	3.326
Mental health issues	0.693	0.165	<0.001	2.001	1.448	2.765
Few social supports	0.488	0.168	0.004	1.629	1.172	2.266
Household Factors						
Overcrowding	0.841	0.337	0.012	2.319	1.199	4.485
2 or more moves in the past year	0.156	0.303	0.606	1.169	0.646	2.116
Ran out of money for basic necessities in the past 6 months *	0.512	0.222	0.021	1.669	1.080	2.580
Case factors						
Previous case openings	−0.044	0.165	0.790	0.957	0.693	1.322
Emotional harm	1.314	0.160	<0.001	3.722	2.718	5.097
Referral source (compared to all other referral sources)						
Medical referral source	1.239	0.272	<0.001	3.452	2.024	5.886

* Basic necessities include food, housing, utilities, telephone, transportation, and medical care.

## Data Availability

The data are not publicly available due to confidentiality concerns and the sensitive nature of the data.

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
