# Peer review of "Child Welfare Investigations of Exposure to Intimate Partner Violence Referred by Medical Professionals in Ontario: A Uniquely Vulnerable Population?"

_healthcare, 2023, doi:10.3390/healthcare11182599_

Round 1

Reviewer 1 Report

1. I recommend that authors use the definition of IPV more recent from WHO and United Nations 

2. I recommend a more in-depth analysis of the studies that have been developed on the subject under study 

3. Regardind materials and methods, I recommend the use of subtitles, for example a subtitle for samples, data analysis, etc

4. Regarding the results and discussion, I recommend that the authors reintroduce the objective and again the use of subtitles 

5. The conclusion is barely as extensive as the discussion. The results need to be discussed in greater depth. Summarize key results with reference to study objectives.

Reviewer 2 Report

It seems to me that the subject of the article is of great academic and practical interest. The analysis of the consequences of who drives the complaints or investigations for Intimate Partner Violence is not a minor issue. If, as this study points out, there are certain "prototypes" of circumstances, depending on who requests the investigation into the welfare of the child suspected of suffering IPV, and there are certain "prototypes" of consequences, this work should compare all sources and not just the medical one. 

Aside from this central issue, the article requires major revisions. First, in the introductory part. There is no review of the texts that have investigated the origin of social services research on child welfare, but above all, there is no clear exposition of all the literature on VCI, its causes and consequences, the origin of its reports, and above all the relationship or influence it has on the welfare of children. On the other hand, there are no hypotheses based on the literature, but some studies are referred to in the discussion. The exposition of the problem and the rationale: why it is necessary to investigate this.

On the other hand, the methodology is very little explained. The reader needs to understand what the database used consists of, how it is put together, what the authors mean when they talk about "research", what type of methodology and data are collected, and exactly what we are talking about when we refer to the sources (complaints, notifications ????...what the system is like in Canada).

The article may be a useful text but requires major revisions. 

Reviewer 3 Report

Dear authors,

The topic analyzed in the article is important and relevant today all over the world.

Regardless of its relevance, I see areas for improvement in the presented article.

In the introductory part of the article, the situation in Canada and Ontario is widely presented, but this part does not include an overview of already conducted research, and does not discuss the analyzed object. Perhaps its purification would help the authors to concentrate on the analyzed problem.

The specific research questions raised by the authors, in my opinion, are more practical than scientific.

The entire purpose of the study is also difficult to grasp.

In the second part "Materials and Methods" Tabel 1 (in line 144) I would suggest either simplifying it or moving it to the annexes part.

In this part, the research subjects are also not clearly presented, there is no information about permission for the study.

Research results are presented clearly.

The discussion part lacks a broader discussion by the authors, which includes the results and conclusions of other studies in the analyzed problem.

The research findings lack specificity and validity

Round 2

Reviewer 2 Report

The text has been greatly improved by the additions the authors have made in response to the reviewers' suggestions. However, there are still two important issues to improve.

All information is limited to Canada. Given that the topic of IPV is transversal, it is suggested that in the introduction references be introduced to studies carried out in other countries regarding the sources of complaints and that these studies be incorporated into the discussion. It is also encouraged to review the writing of the text, which can be improved.

The text has been greatly improved by the additions the authors have made in response to the reviewers' suggestions. However, there are still two important issues to improve.

All information is limited to Canada. Given that the topic of IPV is transversal, it is suggested that in the introduction references be introduced to studies carried out in other countries regarding the sources of complaints and that these studies be incorporated into the discussion. It is also encouraged to review the writing of the text, which can be improved.

Reviewer 3 Report

Dear Authors,

 I hope that the comments allowed  to improve your text. However, it would still be great if you could take into account the note related to the subjects - whether they were really organizations or the relevant persons working in these organizations were interviewed. Perhaps they were the subjects
